# Long-Term Follow-Up after Laser-Assisted Pulmonary Metastasectomy Shows Complete Lung Function Recovery

**DOI:** 10.3390/cancers16091762

**Published:** 2024-05-01

**Authors:** Daniel Baum, Axel Rolle, Dirk Koschel, Lysann Rostock, Rahel Decker, Monika Sombati, Florian Öhme, Till Plönes

**Affiliations:** 1Department of Thoracic Surgery, Fachkrankenhaus Coswig, Lung Center, Neucoswiger Str. 21, 01640 Coswig, Germany; 2Division of Pneumology, Medical Department I, Medical Faculty and University Hospital Carl Gustav Carus, TUD Dresden University of Technology, Fetscherstr. 74, 01307 Dresden, Germany; 3Department of Internal Medicine and Pneumology, Fachkrankenhaus Coswig, Lung Center, Neucoswiger Str. 21, 01640 Coswig, Germany; 4Division of Thoracic Surgery, Department of Visceral, Thoracic and Vascular Surgery, Medical Faculty and University Hospital Carl Gustav Carus, TUD Dresden University of Technology, Fetscherstr. 74, 01307 Dresden, Germany; 5National Center for Tumor Diseases (NCT/UCC Dresden), Fetscherstraße 74, 01307 Dresden, Germany; 6German Cancer Research Center (DKFZ), 69120 Heidelberg, Germany; 7Medical Faculty and University Hospital Carl Gustav Carus, TUD Dresden University of Technology, 01069 Dresden, Germany; 8Helmholtz-Zentrum Dresden—Rossendorf (HZDR), 01328 Dresden, Germany; 9Division of Visceral Surgery, Department of Visceral, Thoracic and Vascular Surgery, Medical Faculty and University Hospital Carl Gustav Carus, TUD Dresden University of Technology, Fetscherstr. 74, 01307 Dresden, Germany

**Keywords:** pulmonary metastasectomy, laser-assisted resection, lung function, forced expiratory volume

## Abstract

**Simple Summary:**

Lung metastases are a major challenge in advanced cancer treatment because they are common and difficult to manage. This study assesses the effectiveness of laser-assisted pulmonary metastasectomy, a surgical technique designed to minimize the removal of healthy lung tissue, thereby preserving lung function. By examining the recovery of lung function in 126 patients after surgery, we aim to fill the gap in long-term data on this procedure. The findings indicate that lung function, measured by the amount of air a person can forcefully exhale in one second, significantly recovers within a year post-surgery. This study demonstrates the benefits of laser-assisted surgery for patients with lung metastases, potentially influencing treatment decisions and improving patient outcomes by maintaining essential lung capacity.

**Abstract:**

Preserving maximum lung function is a fundamental goal of parenchymal-sparing pulmonary laser surgery. Long-term studies for follow-up of lung function after pulmonary laser metastasectomy are lacking. However, a sufficient postoperative lung function is essential for quality of life and reduces potential postoperative complications. In this study, we investigate the extent of loss in lung function following pulmonary laser resection after three, six, and twelve months. We conducted a retrospective analysis using a prospective database of 4595 patients, focusing on 126 patients who underwent unilateral pulmonary laser resection for lung metastases from 1996 to 2022 using a 1318 nm Nd:YAG laser or a high-power pure diode laser. Results show that from these patients, a median of three pulmonary nodules were removed, with 75% presenting central lung lesions and 25% peripheral lesions. The median preoperative FEV_1_ was 98% of the predicted value, decreasing to 71% postoperatively but improving to 90% after three months, 93% after six months, and 96% after twelve months. Statistical analysis using the Friedman test indicated no significant difference in FEV_1_ between preoperative levels and those at six and twelve months post-surgery. The findings confirm that pulmonary laser surgery effectively preserves lung function over time, with patients generally regaining their preoperative lung function within a year, regardless of the metastases’ location.

## 1. Introduction

Pulmonary metastases represent the most common site of metastatic spread in extrapulmonary primary malignancies [1,2,3]. The management of lung metastases poses a significant challenge in clinical practice [4,5]. Local therapy of metastases, especially pulmonary metastasectomy, is a widely accepted therapeutic option [6], capable of significantly prolonging the long-term survival of selected patients [7,8,9,10,11,12]. Indications for pulmonary metastasectomy are technically resectable metastases and primary cancer that is under control or controllable. There should be no uncontrolled or uncontrollable extrathoracic metastasis present. The patient also should have sufficient pulmonary reserve to tolerate the surgery and to maintain quality of life. There should be no alternative medical treatment options available with lower associated morbidity [13].

Laser-assisted metastasis resection offers a safe method to remove even deep-located pulmonary nodes [14]. The preserving of a maximum amount of healthy lung parenchyma was repeatedly used as an important argument for laser resection and against classical wedge resection, even if there was very limited data. The logic behind this is that even centrally located small metastases can be enucleated, and alternatively, anatomical resections or large wedges would have to be performed. However, preserving lung parenchyma may also have implications for potentially enabling further resections since many patients even have to undergo a repeated pulmonary metastasectomy [15,16,17]. Bilateral lung metastases are common, and subsequent metastatic recurrences may occur [18], allowing for repeated resections under the same conditions, with the goal of preserving as much lung parenchyma as possible. However, there is a scarcity of long-term follow-up data demonstrating lung function over an extended period following (initial) pulmonary metastasectomy.

In this study, our primary aim is to comprehensively examine the long-term impact of laser-assisted pulmonary metastasectomy on lung function in patients with pulmonary metastases. Specifically, we seek to quantify the extent of immediate postoperative lung function decline, the subsequent recovery trajectory over a 12-month period, and the ultimate restoration of lung function.

Furthermore, we emphasize the clinical significance of our investigation. By shedding light on the long-term effects of this surgical approach, we aspire to provide clinicians and surgeons with invaluable insights for informed decision-making and patient counseling. Ultimately, this knowledge may contribute to enhanced patient quality of life and facilitate the selection of optimal treatment strategies.

In conclusion, our study endeavors to fill a critical knowledge gap in the realm of pulmonary metastasectomy. We posit that the insights gained from this research will have a meaningful impact on clinical practice and patient care.

## 2. Materials and Methods

### 2.1. Patients

Data from patients with pulmonary metastasis who underwent curative laser-assisted metastasectomy at the Department of Thoracic Surgery, Specialist Hospital Coswig, Saxony, Germany, were collected and stored in a dedicated prospective database. For this retrospective analysis, we reviewed the medical records of patients treated between 1996 and 2022 who met the following inclusion criteria: patients underwent one unilateral pulmonary resection via anterolateral thoracotomy and lung metastases were resected using a 1318 nm laser system. Preoperative pulmonary function thresholds were established to define operability and assess surgical risk. Patients were considered suitable for surgery with an average operative risk if they demonstrated a predicted postoperative FEV_1_ of 40% or higher, DLCO of 40% or higher of the predicted value and an oxygen saturation greater than 90%. Patients participated in a minimum one-year follow-up period, during which lung function assessments were repeatedly conducted using body plethysmography. Exclusion criteria included any additional lung surgeries, especially further metastasectomies, within the specified follow-up period. We also recorded data on the age of patients at the time of metastasectomy, categorized the size of resected nodules, the localization of nodules (peripheral/central), number of resected nodules, primary tumor, and gathered information on the relevant comorbidities and oxygen partial pressure measured in capillary blood.

Following the guidelines of the European Society of Thoracic Surgery, nodules were considered peripheral if they were located within the inner two-thirds of the hemithorax. In fact, various non-comparable definitions exist for these classifications [19].

Patient selection for pulmonary laser resection was guided by a set of inclusion and exclusion criteria aligned with general thoracic surgery standards but also specific to the capabilities and advantages of laser technology. The inclusion criteria were broad, permitting any primary malignancy provided that the primary tumor had been completely resected and there were no incompletely resected extrathoracic metastases. We accepted patients with single or multiple lung metastases, both synchronous and metachronous, and regardless of their count, subject to functional and technical resectability assessments by a thoracic surgeon proficient in laser techniques.

The exclusion criteria reflected a cautious approach to patient well-being and operability: any patient with a Karnofsky performance status of less than 80%, severe cardiac disease, or other high-risk factors were not considered for laser resection.

In the course of our study, spanning from 1996 to 2022, two distinct laser systems, both operating at a wavelength of 1318 nm, were utilized to perform pulmonary laser resection. Initially, the only available technology was the Nd:YAG laser. By 1996, advancements had led to an improved Nd:YAG laser system, optimizing the system’s efficiency, though the process remained time-consuming. In 2007, a significant technological breakthrough was achieved with the development of a high-power pure diode laser, significantly reducing the resection time to one-third of what was previously possible.

From 1996 until the introduction of the diode laser in 2007, the Nd:YAG laser was employed; thereafter, the diode laser became the system of choice. Throughout this study period, the selection of the laser system was determined by its availability and appropriateness for the surgical tasks at hand, ensuring that the most suitable technology was always used to achieve optimal clinical outcomes.

### 2.2. Surgical Technique

The surgical procedure employed for pulmonary metastasectomy involved the following steps:The patient was placed in the lateral decubitus position under general anesthesia, with selective lung ventilation using a double-lumen endotracheal tube.An anterolateral muscle-sparing thoracotomy approach was utilized for unilateral pulmonary resection.Following exploration of the thoracic cavity, metastatic lesions in the lung were identified using visual inspection and systematic palpation. Once located, the metastases were carefully resected using laser technology. The 1318 nm laser system, either a Nd:YAG laser or a high-power pure diode laser, was utilized for precise and controlled tissue ablation. During the laser metastasectomy, meticulous attention was paid to maintaining a precise distance of 2 to 3 mm from the tumor margin. This ensured accurate tumor resection while simultaneously necrotizing a 5 mm wide margin of residual lung tissue due to the dispersal of laser energy, thereby optimizing resection and preserving surrounding tissue. Care was taken to ensure the complete removal of all visible metastatic lesions while preserving healthy lung parenchyma. Following the resection of lung metastasis, the lung parenchyma is reconstructed by carefully re-approximating the visceral pleura using absorbable sutures. Additionally, a systematic lymphadenectomy was performed as a standard procedure.Adequate hemostasis was achieved using electrocautery and suturing techniques as necessary. Chest tubes were inserted, and the thoracic cavity was carefully closed layer by layer.Following surgery, patients were closely monitored in the intensive care unit (ICU) and subsequently transferred to the surgical ward. Regular assessments of vital signs, pain management, and respiratory function were conducted. Patients received appropriate postoperative care; comprehensive physiotherapy began on the day of surgery. This included early mobilization with support from physiotherapists. Additionally, patients were instructed in the use of an incentive spirometer, which they were required to use a minimum of six times per hour post-surgery to facilitate slow and deep inhalations. The device provided visual feedback to track their progress. The physiotherapy regimen also incorporated kinesiotherapy and specialized respiratory exercises, conducted either individually or in group sessions. They were advised to continue using the spirometer for six weeks after discharge, although this was not monitored.

### 2.3. Pulmonary Function Tests

For the lung function tests, body plethysmography was performed in our pulmonary function department by professionally trained MTAs according to the current guidelines of ERS/ATS. Lung function parameters included forced expiratory volume in one second (FEV_1_), forced vital capacity (FVC), vital capacity (VC), residual capacity (RV), diffusing capacity of the lung for carbon monoxide (DLCO), and total lung capacity (TLC). Each parameter was specified in percentage (%). Pulmonary function was measured prior to surgery, before discharge from the hospital, after three months, after six months, and after twelve months. Subsequently, patients also were followed up at our outpatient clinic, where regular CT scans were performed.

### 2.4. Statistics

Statistical analysis was conducted using MedCalc^®^ Statistical Software version 20.104 (MedCalc Software Ltd., Ostend, Belgium). Categorical and count data were expressed as frequencies and percentages. Given the non-normal distribution of these data sets, the Friedman test was performed for analyzing repeated measurements. Statistical significance was defined as a *p*-value less than 0.05.

## 3. Results

A total of 126 surgical procedures were analyzed in 126 patients, all of which were conducted via anterolateral thoracotomy. The median age of patients at the time of surgery was 64.3 years, with an age range spanning from 20 to 79 years. Among the patients, 58 (46%) were male. Each surgical intervention involved the resection of a median of three metastases, with the number of metastases per operation ranging from 1 to 16. Furthermore, it was observed that 75% of all patients presented with central lung lesions, and 25% had only peripheral lung lesions. Systematic lymphadenectomy was performed in all surgical procedures. There were no recorded instances of perioperative mortality. Anatomical resections or wedge resections were not performed during any of the surgeries. In this cohort of 126 patients, postoperative complications classified according to the Clavien–Dindo scale were as follows: Grade I in 3.15% (*n* = 4), Grade II in 11.02% (*n* = 14), and Grade IIIb in 0.79% (*n* = 1) due to a re-thoracotomy under general anesthesia caused by bleeding; Grade IV in 1.57% of cases (*n* = 2) attributed to postoperative acute myocardial infarction (AMI) and stroke. Specific complications included postoperative bleeding (0.79%), air leakage in five patients (3.94%), and pneumonia in seven patients (5.51%). Other complications encompassed respiratory insufficiency, as well as less frequent cardiac and intestinal complications. The histological composition of the metastases consisted of renal cell carcinoma in 21% of cases, colorectal carcinoma in 21%, breast carcinoma in 12%, and various other malignancies, including urogenital carcinoma, melanoma, carcinoma of the ear, nose, and throat (ENT), fibrous histiocytoma, esophageal carcinoma, prostate carcinoma, sarcoma, thyroid carcinoma, seminoma and carcinoma of unknown primary (CUP). Detailed clinical characteristics are presented in Table 1.

Preoperative lung function parameters for the entire cohort revealed nearly normal respiratory function, with a median FEV_1_ in percent predicted of 98% (range from 58% to 139%). Postoperative lung function analysis, conducted on average 1 day prior to discharge from the hospital, demonstrated a median FEV_1_ of 71%; ∆ = 27% (range from 36% to 122%); *p* < 0.001. At 3 months postoperation, the median FEV_1_ was 90%; ∆ = 8% (range from 45% to 128%). After 6 months, 93% ∆ = 5% (range from 58% to 136%). After 12 months, the median FEV_1_ in percent predicted was 96% ∆ = 2% (range from 44% to 132%), all *p* > 0.05. (See Figure 1).

Patients with only peripheral nodules at the time of surgery demonstrated normal respiratory function, with a median FEV_1_ in percent predicted of 99% (ranging from 69% to 127%). Postoperative lung function analysis for patients with resected peripheral nodules, conducted on average one day prior to discharge from the hospital, revealed a median FEV_1_ of 73%; ∆ = 26% (range from 36% to 111%); *p* < 0.001. At 3 months postoperation, the median FEV_1_ was 94%; ∆ = 5% (range from 58% to 122%). After 6 months, it was 96%; ∆ = 3% (range from 58% to 136%). After 12 months, the median FEV_1_ in percent predicted was 91% ∆ = 8% (range from 55% to 130%), all *p* > 0.05 (See Figure 2a).

Preoperative lung function parameters for patients with central nodules showed unaltered respiratory function, with a median FEV_1_ in percent predicted of 107% (range from 69% to 128%). Postoperative lung function analysis for patients with central nodules, conducted on average one day prior to discharge from the hospital, revealed a median FEV_1_ of 78%; ∆ = 29% (range from 46% to 105%); *p* < 0.001. At 3 months postoperation, the median FEV_1_ was 94%; ∆ = 13% (range from 63% to 128%). After 6 months, it was 97%; ∆ = 10% (range from 64% to 124%). After 12 months, the median FEV_1_ in percent predicted was 102% ∆ = 5% (range from 54% to 123%), all *p* > 0.05 (See Figure 2b).

A comparative analysis was performed to evaluate the pulmonary functional changes post-surgery between patients with central and peripheral lung metastases. The changes in FEV_1_ were assessed using Analysis of Variance (ANOVA). The statistical analysis revealed no significant differences in the recovery of pulmonary function between the groups (*p* > 0.05). These results indicate that the location of metastases, whether central or peripheral, does not significantly affect the outcomes of pulmonary function. Both patient groups demonstrated comparable improvements in lung function following the surgical intervention, affirming the effectiveness of the procedure irrespective of the metastatic site within the lung.

## 4. Discussion

In this study, we investigated the impact of laser-assisted pulmonary metastasectomy on long-term lung function in patients with pulmonary metastases. Our findings unveil several critical insights:

Laser-assisted pulmonary metastasectomy is a secure modality for the removal of pulmonary metastases, devoid of any observed mortality [20]. The immediate postoperative decline in lung function after laser-assisted metastasectomy aligns with the outcomes reported in earlier investigations involving conventional methods of pulmonary metastasectomy. Notably, a substantial recovery in lung function is evident after a 3-month period. Furthermore, over time, postoperative lung function approximates preoperative baseline values. Consequently, there is an overall absence of lung function loss with respect to FEV_1_.

In the scope of this study, only patients undergoing a single unilateral metastasectomy were considered. The opportunity for a subsequent intervention on the contralateral side was typically offered 4–6 weeks postoperatively, contingent upon the patient’s recovery from the initial surgery. If the lung was the sole organ affected by metastases, additional chemotherapy was generally not utilized; instead, patients were closely monitored through a rigorous follow-up protocol. It is important to note that the timing of subsequent systemic adjuvant therapy or, for example, mediastinal radiation could proceed without delay, comparable with the protocols followed after anatomical resection for lung carcinoma.

Pioneering work in the field of pulmonary laser resection was carried out by Rolle et al., who initially demonstrated the superiority of the 1318 nm wavelength in comparison with the 1068 nm laser for lung resection [20]. Several studies have also shown that laser resection allows for the removal of a greater number of metastases [21]. Furthermore, Rolle et al. demonstrated that laser resection led to the resection of more metastases, even in cases of multiple bilateral or centrally located metastases. This approach had a significant influence on the preservation of pulmonary parenchyma and appeared to minimize complications [22]. Chandrana et al. showed that significantly less lung parenchyma is resected using a laser compared with traditional stapled wedge resection [23]. Furthermore, it is worth noting that even when employing a monopolar cutter, greater damage to the lung parenchyma is observed compared with laser application in porcine lungs, suggesting a potential advantage of laser techniques over monopolar instruments in preserving lung tissue integrity [24]. All of this leads to the question of whether laser metastasectomy can indeed preserve lung function significantly. In 2009, Petrella et al. demonstrated that after wedge resection of an average of three metastases, there was a mean loss of 13.4% in FEV_1_ after metastasectomy. The authors concluded that the functional loss occurring after three or more non-anatomic resections was comparable with the functional reductions observed after lobectomy [25]. In a prospective study, Welter et al. showed an average loss of lung function, specifically in FEV_1_, of 9.2% at 3 months postoperation following a median of 3 metastasis resections. The results indicated that the loss of FEV_1_ was dependent on the number of wedge resections, with each additional resection resulting in a reduction of FEV_1_ by 23 mL (0.6%). Additionally, there was a decrease of approximately 6% attributed to thoracotomy [26]. Hassan et al. demonstrated that lung function remained at 11% below the preoperative FEV_1_ level after 3 months of postoperative laser metastasectomy (with a median of 2 metastases resected), and there was no significant further recovery after 6 months [15]. None of these studies provided data for long-term follow-up. In contrast, Mori et al. demonstrated that in their study involving 29 patients undergoing lung wedge resection, there was a temporary reduction in forced expiratory volume in one second (FEV_1_) observed at the 3-month postoperative assessment. This was followed by a gradual recovery, although it is noteworthy that FEV_1_ values remained slightly below the preoperative levels even at the 12-month follow-up point [27].

Given that these findings did not align with our clinical experience, and we continued to deal with this question in our everyday practice, we conducted this study to investigate the long-term effects of laser metastasectomy on the lung function of our patients. Our hypothesis was that there would be no clinically significant loss of lung function in the long term.

Following an expected immediate loss of lung function, recovery occurs gradually. Our findings demonstrate that lung function fully recuperates over the long term following open resection of unilateral solitary or multiple metastases, regardless of whether the metastases were centrally or peripherally located. By utilizing the 1318 nm laser, we are ultimately able to preserve a maximum amount of lung tissue since the resection takes place very close to the surface of the metastasis. See Figure 3, which offers a series of illustrative CT images capturing the tissue recovery throughout the postoperative course.

To our knowledge, there is currently no data available on the long-term (>6 months) follow-up assessment of lung function following pulmonary laser metastasectomy.

Our findings demonstrate that pulmonary metastasectomy represents a safe and lung-function-preserving approach to managing pulmonary metastases. However, it is crucial to recognize the evolving dynamics in the field of metastasectomy. These strategies underscore the necessity of exploring novel methodologies within the realm of thoracic surgical metastasectomy, with the potential to augment treatment outcomes. Therefore, it is imperative that we maintain thoracic surgical metastasectomy as a secure component within our thoracic surgical repertoire and thoroughly investigate its impact on our patients.

Furthermore, it is essential to delve into how lung function evolves in patients following bilateral or multiple metastasis resections. This necessitates further research efforts.

The debate over the optimal surgical approach for pulmonary metastases—open thoracotomy versus video-assisted thoracoscopic surgery (VATS)—requires a nuanced consideration that takes into account both advancements in imaging diagnostics and the importance of surgical precision and access. Studies by Eckardt and Licht, as well as Cerfolio et al., have highlighted the advantages of open thoracotomy, especially in detecting and resecting non-imaged malignant nodules through direct manual palpation. This capability for comprehensive examination and treatment of the lung becomes critically important when a large number of metastases are present, as it increases the likelihood of discovering and resecting non-imaged nodules [20,28].

Conversely, this study by Nakas et al. and the more recent research by Prenafeta Claramunt et al. provide valuable insights into the effectiveness and reliability of VATS, supported by modern imaging techniques, which can achieve comparable results to open procedures in selected cases. Prenafeta Claramunt et al.’s findings specifically indicate no significant difference in ipsilateral recurrence rates between VATS and open surgery for colorectal cancer lung metastases, suggesting that the less invasive VATS could be a viable option without compromising on the rate of cancer recurrence [29,30].

Regardless of the chosen surgical approach, when performing an open thoracotomy, it is crucial to treat the lung parenchyma as delicately as possible. This approach can be described as ‘minimally resective,’ which becomes increasingly important as the number of metastases to be removed rises. Such tissue-sparing resection, as we have demonstrated, can enhance postoperative lung function, thereby improving the quality of life for patients.

## 5. Conclusions

In summary, our investigation provides comprehensive insights into the long-term repercussions of laser-assisted pulmonary metastasectomy on lung function among individuals with pulmonary metastases. Our examination reveals that this surgical approach yields no enduring detriment to lung function, even when dealing with multiple and centrally located metastatic lesions. Despite a transient decline in the immediate postoperative phase, lung function experiences a gradual recuperation over time, eventually reinstating itself to baseline values.

These findings underscore the clinical significance of laser-assisted pulmonary metastasectomy as a viable therapeutic option. By clarifying the complex dynamics of lung function recovery following resection, our study equips medical professionals and surgeons with vital knowledge for informed clinical decision-making and patient counseling. Ultimately, this understanding enhances patient quality of life and assists in the selection of optimal treatment strategies.

### Strengths and Limitations

Our study possesses several notable strengths, including a substantial dataset of patient records collected over a significant time frame, the use of standardized surgical techniques, and rigorous follow-up procedures. However, it is important to acknowledge certain limitations:

While conducting this retrospective analysis, we encountered several biases and constraints typical of such studies. Data were collected from a dedicated prospective database and medical records spanning over a period of 26 years, inherently introducing historical bias due to changes in medical practice, technology, and patient management protocols over time.

Selection bias is evident, as this study only included patients who underwent unilateral pulmonary resection via anterolateral thoracotomy and those who completed a minimum one-year follow-up. Consequently, the results may not be generalizable to all patients with pulmonary metastases, especially those with bilateral lesions.

Additionally, the exclusion of patients who underwent further lung surgeries within the follow-up period could also introduce attrition bias, potentially excluding more complex cases with recurrent metastases, thus affecting this study’s generalizability.

The definition of nodule localization as peripheral or central is also subject to interpretation, and non-comparable definitions for these classifications may cause classification bias. All surgical procedures were conducted using a consistent technique; however, intraoperative decision-making and the extent of lymphadenectomy might have varied between surgeons, introducing performance bias.

Furthermore, lung function parameters were measured according to current guidelines, but given the long timespan of this study, changes in these guidelines and measurement technologies could have introduced variability.

Analytical constraints also played a role. The statistical methods employed to handle non-normally distributed data, such as the Friedman test, are appropriate for the data at hand but may not capture all underlying patterns or relationships that more sophisticated models, such as mixed-effects models or multivariate regression, could reveal. Our decision to use non-parametric tests was guided by the distribution of our data, yet this choice may limit the depth of conclusions that can be drawn.

In conclusion, while every effort was made to minimize these biases and constraints, they should be considered when interpreting the findings of our study. Further research is needed, potentially incorporating a more diverse patient cohort and a multi-center design to mitigate these biases and expand the applicability of the findings.

Unfortunately, we lack comprehensive data regarding postoperative pain intensity assessed on an analog pain scale, as well as information on the consumption of analgesics among the studied patients. The collection of such data could have provided additional insights into the postoperative course and potentially unveiled associations between pain levels, analgesic consumption, and the development of lung function.

## Figures and Tables

**Figure 1 cancers-16-01762-f001:**
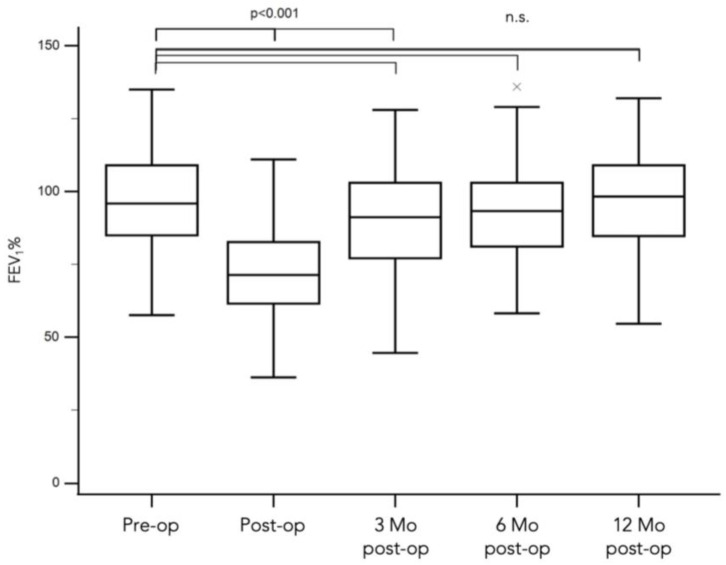
Longitudinal Changes in FEV_1_ Following Laser-Assisted Pulmonary Metastasectomy. Preoperative lung function parameters for the entire cohort were unimpaired, and the median FEV_1_ in percent predicted was 98% (range from 58% to 139%). Immediately performed postoperative lung function analysis was reduced with a median FEV_1_ in percent predicted of 71%; ∆ = 27% (range from 36% to 122%); *p* < 0.001. In follow-up at 3 months postoperation, the median FEV_1_ was 90%; ∆ = 8% (range from 45% to 128%); after 6 months, 93%; ∆ = 5% (range from 58% to 136%) and after 12 months the median FEV_1_ in percent predicted was 96% ∆ = 2% (range from 44% to 132%). These follow-up FEV_1_ recovered and were not significantly altered compared with the preoperative lung function (*p* > 0.05, denoted as n.s. for ‘not significant’).

**Figure 2 cancers-16-01762-f002:**
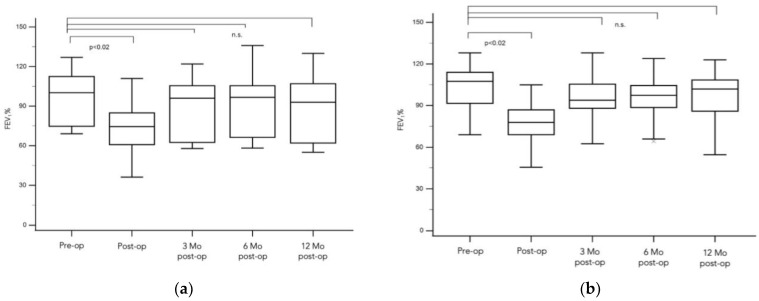
Course of lung function differentiated according to the localization of the metastases. (**a**): Patients with only peripheral nodules at the time of surgery demonstrated unaltered respiratory function, with a median FEV_1_ in percent predicted of 99% (ranging from 69% to 127%). Postoperative lung function analysis for patients with resected peripheral nodules conducted one day prior to discharge from the hospital revealed a slightly impaired FEV_1_ (median of 73%; ∆ = 26% (range from 36% to 111%), whereas three months postoperation, the median FEV_1_ was 94%; ∆ = 5% (range from 58% to 122%). After 6 months and 12 months, lung function recovered, and FEV_1_ was not statistically different from the preoperative FEV_1_ (at 3 months: FEV1 96%; ∆ = 3% (range from 58% to 136%) and after 12 months, the median FEV_1_ in percent predicted was 91% ∆ = 8% (range from 55% to 130%) all *p* > 0.05, denoted as n.s. for ‘not significant’. (**b**): The course of lung function in patients with centrally located nodules was similar to the patients with peripheral metastases. Preoperative lung function parameters showed unaltered respiratory function, with a median FEV_1_ in percent predicted of 107% (range from 69% to 128%), whereas directly measured postoperative lung function analysis revealed a median FEV_1_ of 78%; ∆ = 29% (range from 46% to 105%); *p* < 0.001. Lung function measured after three months postoperation showed a median FEV_1_ of 94%; ∆ = 13% (range from 63% to 128%). After 6 months and 12 months, lung function recovered and returned to the baseline (FEV_1_ 97%; ∆ = 10% (range from 64% to 124% after three months and FEV_1_ 102% ∆ = 5% (range from 54% to 123% after 12 months) all *p* > 0.05, denoted as n.s. for ‘not significant’).

**Figure 3 cancers-16-01762-f003:**
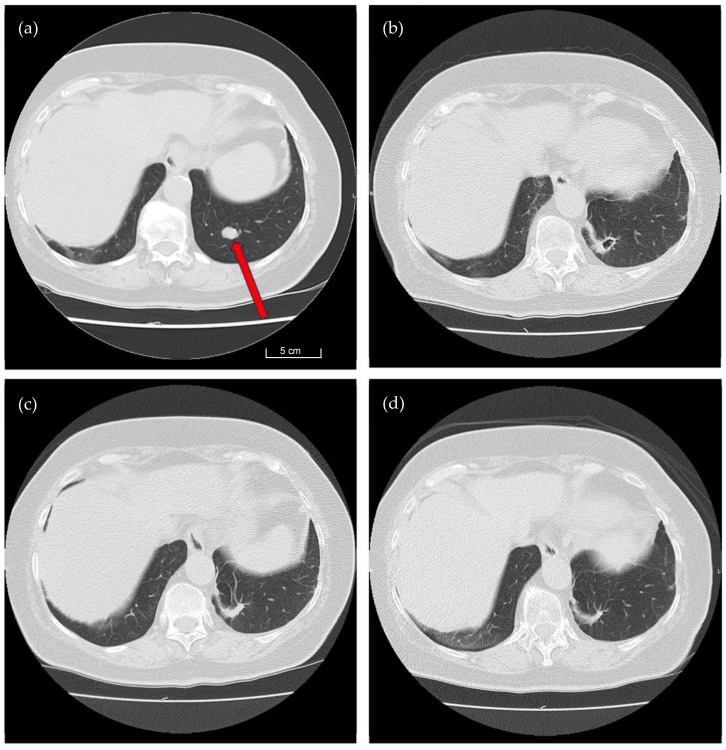
CT Findings Prior and After Laser-Assisted Pulmonary Metastasectomy. This set of four CT images illustrates the longitudinal course of pulmonary metastasectomy in a 74-year-old female patient diagnosed with two metastases of renal cell carcinoma in the left lung. The images provide a comprehensive view of postoperative changes over a period of one year. (**a**): this CT scan was captured four weeks before the laser-assisted metastasectomy. It displays a distinct metastatic lesion measuring approximately 12 mm in Segment 10 of the left lung (red arrow). (**b**): three months after the surgical procedure, Image b reveals significant changes in the left lung. A characteristic air-filled cavity resulting from laser ablation is clearly visible in Segment 10. (**c**): Six months post-surgery, the previously formed pulmonary cavity has completely resolved, leaving only residual scar tissue at the resection site. (**d**): a CT scan captured 12 months postoperatively shows further regression. The residual scar tissue in Segment 10 continues to diminish, demonstrating gradual but progressive recovery.

**Table 1 cancers-16-01762-t001:** Patients and clinical characteristics. This table provides an overview of the demographic and clinical characteristics of the patient cohort undergoing laser metastasectomy and details regarding the pulmonary lesions.

Patients Characteristics	Count	Range/Percentage
Age at metastasectomy, years	64.3 (median)	range from 20–79
Male gender	58	46%
Preoperative existing lung disease	25	20%
chronic obstructive pulmonary disease	19	15%
bronchial asthma	3	2%
condition after pulmonary tuberculosis (PTB)	1	1%
silicosis	2	1.6%
Number of lesions	3 (median)	range from 1–16
Primary malignancy		
renal cell carcinoma	27	21%
colorectal carcinoma	26	21%
urogenital carcinoma	14	11%
breast carcinoma	12	9%
melanoma	6	5%
carcinoma of the ENT	4	3%
bronchial carcinoma	2	1.6%
Central lesions	95	75%
Peripher lesions	31	25%
Number of resected lesions		
1–3	74	59%
4–5	32	25%
6–10	12	9%
11–16	6	5%

## Data Availability

Data presented in this study are available on request from the corresponding author due to concerns about the privacy of personal data.

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
