# Peer review of "Long-Term Follow-Up after Laser-Assisted Pulmonary Metastasectomy Shows Complete Lung Function Recovery"

_cancers, 2024, doi:10.3390/cancers16091762_

Round 1

Reviewer 1 Report

Comments and Suggestions for Authors

Congratulations to you on your successful lung metastasectomy using laser as an effective tool. It is quite surprised that after the resection, the pulmonary function could even recover to its pre-operative level. It is encouraging. There are some questions which were not mentioned in the text, should be addressed before consideration of publication.

(1) By how far from the margin of the tumor were your laser beam applied ?

(2) Were the patients received  pulmonary physiotherapy after the operation ?

(3)The postopeartive complications were not  mentioned.

(4) Did you compare the post-operative pulmonary functional change between central  and peripheral cases ?

(5) Were there any bilateral cases ?  If there were, what were your treatment strategies, staged or simultaneous operation ? And how were the pulmonary functional change for those bilateral cases ?

Reviewer 2 Report

Comments and Suggestions for Authors

The authors investigated the impact of laser-assisted pulmonary metastasectomy on long term lung function in patients with pulmonary metastases with average age of 64.3 years. The patients had different types of cancer and the histological composition of the metastases consisted of renal cell carcinoma , colorectal carcinoma, breast carcinoma, and various other malignancies, like urogenital carcinoma, melanoma, carcinoma of the ear, nose, and throat (ENT), fibrous histiocytoma, esophageal carcinoma, prostate carcinoma, sarcoma, thyroid carcinoma, seminoma and carcinoma of unknown primary (CUP). The lung function parameters recorded were forced expiratory volume in one second (FEV1), forced vital capacity (FVC), vital capacity (VC), residual capacity (RV), diffusing capacity of the lung for carbon monoxide (DLCO) and total lung capacity (TLC). Pulmonary function was measured prior to surgery, before discharge of hospital, after three months, after six months and after twelve months. A total of 126 surgeries were done with number of metastases per operation ranging from 1 to 16 using utilizing the 1318 nm laser. Their examination reveals that this surgical approach yields no enduring detriment to lung function, even when dealing with multiple and centrally located metastatic lesions. Despite a transient decline in the immediate postoperative phase, lung function experiences a gradual recuperation over time, eventually reinstating itself to baseline values. The research done was very impressive yielding a very positive response towards the patients undergoing lung metastasis. I have a few minor comments mentioned below:

1.      Before taken for surgery, was there any threshold value of PFT which was to be maintained?

2.      Post surgery, what were the drugs used for faster healing of the laser affected wound?

3.      Post surgery how much time was needed to recover and be ready for the next round of chemotherapy or targeted therapy or radiation?

These parameters may vary from patient to patient, but I think it can be added as a small paragraph of discussion in the revised manuscript.

I recommend minor revision.

Reviewer 3 Report

Comments and Suggestions for Authors

The study explores the long-term impact of pulmonary laser resection on lung function among patients with lung metastases. Conducted as a retrospective analysis, the study scrutinized data from 126 patients who underwent unilateral pulmonary laser resection, utilizing either a 1318 nm Nd:YAG laser or a high-power pure diode laser, spanning from 1996 to 2022. Results reveal an initial decrease in lung function postoperatively, with patients generally recovering their preoperative lung function within a year. This suggests the efficacy of pulmonary laser surgery in preserving lung function over time, irrespective of the metastases' location. The manuscript is well written, posing a well-defined question and adhering to reporting standards.

To enhance the manuscript, the authors should address several points.

The authors should explain the biases and constraints inherent in data collection and analysis.

The authors need to provide information on how patients were selected for pulmonary laser resection or how the choice of laser type was determined. Clear criteria for patient selection and treatment allocation should be provided to assess potential selection bias.

While the Friedman test was used to analyze changes in FEV1 over time, other statistical methods, such as mixed-effects modeling or repeated measures ANOVA, could provide a more robust analysis of longitudinal data. Additionally, considering the small sample size, the power of the statistical tests should be discussed.
